# ATF3 downmodulates its new targets IFI6 and IFI27 to suppress the growth and migration of tongue squamous cell carcinoma cells

Lin Xu[1,2,3,4], Tingjian Zu[1,5], Tao Li[1,2], Min Li[4], Jun Mi[1], Fuxiang Bai[1], Guanyi Liu[1], Jie Wen[1], Hui Li[6], Cord Brakebusch[7], Xuxia Wang[2]*, Xunwei Wu[1]*

1 Department of Tissue Engineering and Regeneration, School and Hospital of Stomatology, Cheeloo College of Medicine, Shandong University & Shandong Key Laboratory of Oral Tissue Regeneration and Shandong Engineering Laboratory for Dental Materials and Oral Tissue Regeneration, Jinan, Shandong, China, 2 Department of Oral and Maxillofacial Surgery, School and Hospital of Stomatology, Cheeloo College of Medicine, Shandong University & Shandong Key Laboratory of Oral Tissue Regeneration & Shandong Engineering Laboratory for Dental Materials and Oral Tissue Regeneration, Shandong, China, 3 Department of Orthodontics, Liaocheng People's Hospital, Liaocheng, Shandong, China, 4 Precision Biomedical Key Laboratory, Liaocheng People's Hospital, Liaocheng, Shandong, China, 5 School of Stomatology, Shandong First Medical University & Shandong Academy of Medical Sciences, Tai'an, Shandong, China, 6 Department of Hematology, Southwest Hospital, Third Military Medical University, Chongqing, China, 7 Biotech Research and Innovation Centre (BRIC), University of Copenhagen, Ole Maaløes Vej 5, Copenhagen, Denmark

* wxx@sdu.edu.cn (XW); xunwei_2006@hotmail.com (XW)

## Abstract

Activating transcription factor 3 (ATF) is a key transcription factor involved in regulating cellular stress responses, with different expression levels and functions in different tissues. ATF3 has also been shown to play crucial roles in regulating tumor development and progression, however its potential role in oral squamous cell carcinomas has not been fully explored. In this study, we examined biopsies of tongue squamous cell carcinomas (TSCCs) and found that the nuclear expression level of ATF3 correlated negatively with the differentiation status of TSCCs, which was validated by analysis of the ATGC database. By using gain- or loss- of function analyses of ATF3 in four different TSCC cell lines, we demonstrated that ATF3 negatively regulates the growth and migration of human TSCC cells *in vitro*. RNA-seq analysis identified two new downstream targets of ATF3, interferon alpha inducible proteins 6 (IFI6) and 27 (IFI27), which were upregulated in ATF3-deleted cells and were downregulated in ATF3-overexpressing cells. Chromatin immunoprecipitation assays showed that ATF3 binds the promoter regions of the IFI6 and IFI27 genes. Both IFI6 and IFI27 were highly expressed in TSCC biopsies and knockdown of either IFI6 or IFI27 in TSCC cells blocked the cell growth and migration induced by the deletion of ATF3. Conversely, overexpression of either IFI6 or IFI27 counteracted the inhibition of TSCC cell growth and migration induced by the overexpression of ATF3. Finally, an *in vivo* study in mice confirmed those *in vitro* findings. Our study suggests that ATF3 plays an anti-tumor function in TSCCs through the negative regulation of its downstream targets, IFI6 and IFI27.

**Data Availability Statement:** The data that support the findings of this study are available from Dr. Qun Zhang (email: zhangqunpku@126.com) upon reasonable request. The RNA-seq data have been

submitted to the NCBI database under Submission ID: SUB7107591 with the accession code: PRJNA610975.

**Funding:** This work was supported by The National Key Research and Development Program of China (2017YFA0104604) (http://www.most.gov.cn), the General Program of National Natural Science Foundation of China (81772093, 82073470) (https://isisn.nsfc.gov.cn), the Key Program of Shandong Province Natural Science Foundation (ZR2019ZD36) and The Key Research and Development Program of Shandong Province (2019GSF108107) (http://cloud.sdstc.gov.cn/). All funds were awarded to X.WU. The funders had no role in study design, data collection and analysis, decision to publish, or preparation of the manuscript.

**Competing interests:** The authors have declared that no competing interests exist.

## Author summary

Activating transcription factor 3 (ATF3), a stress response gene, has been shown to play either tumor promoting or tumor suppressing functions depending on the type of tumor cell and the stromal context. Here we discovered that ATF3 plays an anti-tumor role in tongue squamous cell carcinoma (TSCC) cells through the transcriptional suppression of its new downstream targets interferon alpha inducible proteins 6 (IFI6) and 27 (IFI27). This finding contributes to understanding how ATF3, a transcriptional repressor, can target specific downstream genes in different tumor cells to play anti-tumor or pro-tumor functions. A thorough understanding of ATF3 functions and its downstream signaling pathways provides a potential approach to develop new therapeutics for the treatment of tumors such as TSCCs.

## Introduction

Head and neck squamous cell carcinomas (HNSCCs) are one of the top ten most fatal cancers in the world. Twenty-five % of HNSCCs are oral squamous cell carcinomas (OSCCs) [1,2] of which nearly half are tongue squamous cell carcinomas (TSCCs). TSCCs are highly aggressive tumors that are prone to local recurrence and metastasis and are difficult to treat as indicated by a five-year survival rate of only 50% [3]. Many factors, including genetic predisposition, lifestyle (such as alcohol and tobacco consumption) and viral infection (human papillomavirus), are associated with the risk for TSCCs, however the detailed molecular mechanism involved in TSCC development is unclear [3,4]. More knowledge of the molecular mechanisms underlying TSCC development is therefore urgently required as a basis for new therapeutic strategies for the clinical treatment of TSCCs.

ATF3 (activating transcription factor 3), a member of the CREB (ATF/cyclic AMP-responsive element binding protein) family, is a key transcription factor involved in regulating cellular stress responses with different functions in different tissues [5,6]. The roles of ATF3 in tumor cells have been intensively investigated in recent years [6,7]. Interestingly, ATF3 can act as an oncogene or as a tumor suppressor gene, depending on the type of tumor and the cellular context [8–12]. For instance, ATF3 promotes tumor cell growth in skin, breast and lung cancer cells [13–16]. In contrast, ATF3 was found to be down-regulated in esophageal squamous cell carcinomas, hepatocellular cancers, prostate and colon cancers and thus acts as a tumor suppressor [17–22]. Recent studies have identified ATF3 as a downstream target of microRNAs miR-488 and miR-431, which play important roles in the progression of TSCCs [23,24]. However, the downstream effects of ATF3 in TSCCs have not been explored so far.

The present study investigated the role of ATF3 in the growth and migration of TSCC cells and explored its underlying molecular mechanism. The results demonstrate that ATF3 plays a tumor suppressing function in TSCCs through the negative regulation of its novel targets IFI6 and IFI27.

## Results

### Lower levels of ATF3 are expressed in poorly differentiated TSCCs derived from clinical patients

In order to characterize the potential role(s) of ATF3 in TSCCs, the expression of ATF3 was analyzed by immunohistochemistry (IHC) in highly (n = 11), moderately (n = 8) and poorly differentiated TSCCs (n = 10), as well as in normal tongue tissues (n = 6), graded previously by histological analysis (Fig 1A). Immunohistochemical staining of ATF3 was carried out in those tumors and representative images of ATF3 staining in 3 different tissues (Cases 1, 2 and 3)

from each group are shown in S1 Fig. One of them (Case 1 from each group) are shown at high magnification in Fig 1B. From the staining patterns visible in Figs 1B and S1, we can observe that ATF3 is localized in the nuclei of normal oral epithelial(E) cells but nuclear staining of ATF3 is decreased in poorly differentiated TSCCs (Figs 1B (red arrows) and S1). Interestingly, the cytoplasmic staining of ATF3 in TSCC cells, especially in highly differentiated TSCC cells, was also strong, suggesting there is a potential redistribution of ATF3 between the nucleus and cytoplasm during tumorigenesis and malignant progression (Figs 1B and S1). These results indicate that ATF3 has various expression patterns in different OSCCs. Therefore, we assessed the ATF3 mRNA expression levels in 5 oral SCC lines, and found that two of those 5 cell lines expressly relative high levels of ATF3 (CAL 27 and SCC-25) compared to the other three cell lines (SCC-4, SCC-9 and SCC022) (S2 Fig). We then stained ATF3 in CAL 27 and in SCC-9 cells as well as in normal oral derived keratinocytes (OHKCs) and found a significantly higher nuclear staining of ATF3, which colocalized with nuclear staining (DAPI, blue), in normal oral epithelial cells and slightly higher levels of ATF3 in the nuclei of CAL 27 cells compared to SCC-9 cells (red arrows, Fig 1C and individual images of ATF3 and DAPI staining are shown in S3 Fig). The quantification of cells that were positive for ATF3 in their nuclei in Fig 1B and 1C are shown in Fig 1D and 1E, respectively, indicated that the percentage of cells positive for ATF3 in their nuclei was significantly higher in normal oral epithelial cells compared to tumor cells (Fig 1D and 1E). That result was validated by western blot analysis of ATF3 protein levels in nuclear fractions prepared from cultured epithelial and TSCC cells (Fig 1F). To further confirm the lower expression of ATF3 in oral SCCs, we analyzed ATF3 expression in HNSCCs in the TCGA dataset using the UALCAN tool (http://ualcan.path.uab.edu) [25]. That analysis revealed a slightly lower expression level of ATF3 in primary tumors compared with normal tissues (S4 Fig), although the difference was not statistically significant. However, when we selected only TSCCs in the TCGA dataset with NIH GDC portal for analysis of ATF3 expression level (https://portal.gdc.cancer.gov), significantly lower mRNA expression levels of ATF3 were found in oral SCCs compared to normal tissues (Fig 1G).

These data suggest that the expression of ATF3, especially ATF3 localized in nuclei, is downregulated in TSCCs, which suggests that ATF3 may play a tumor suppressing function in TSCC cells.

## Loss of ATF3 promotes TSCC cell growth and migration

To further understand the role of ATF3 in TSCCs, we deleted the ATF3 gene in CAL 27 cells, which express relatively high levels of ATF3 (S2 Fig) using CRISPR genome editing. In order to validate the effect of ATF3 deletion, we designed two independent small guide RNAs (sgRNA1 and sgRNA2) to create frame-shift mutations in ATF3. Those sgRNAs resulted in the expression of a truncated form of ATF3 by sgRNA1 which specifically targeting the end region of the ATF3 exon right before the 3'-UTR, or in an efficient loss of ATF3 by sgRNA2, which targeting the end of the 5'-UTR of the ATF3 gene (S5A Fig). Both genome edited populations showed significantly increased growth compared to cells transfected with an empty vector (V) starting at day 3 after infection (Fig 2A). We didn't observe a further increase of cell growth after day 3, which could result from an unknown feedback mechanism due to the deletion of ATF3 in vitro. To verify this result, we inhibited ATF3 expression using the transfection of specific siRNAs, whose knockdown efficiencies were reported in our previous publication [13] and were also validated by qRT-PCR (S6A Fig) in these cells and produced the same phenotype (S6B Fig). Furthermore, cell migration was increased by sgRNA1 and by sgRNA2 as revealed by trans-well migration assays (Fig 2B and 2C) and by wound healing assays (Fig 2D and 2E). These results were further validated using another TSCC line SCC-25 which has a

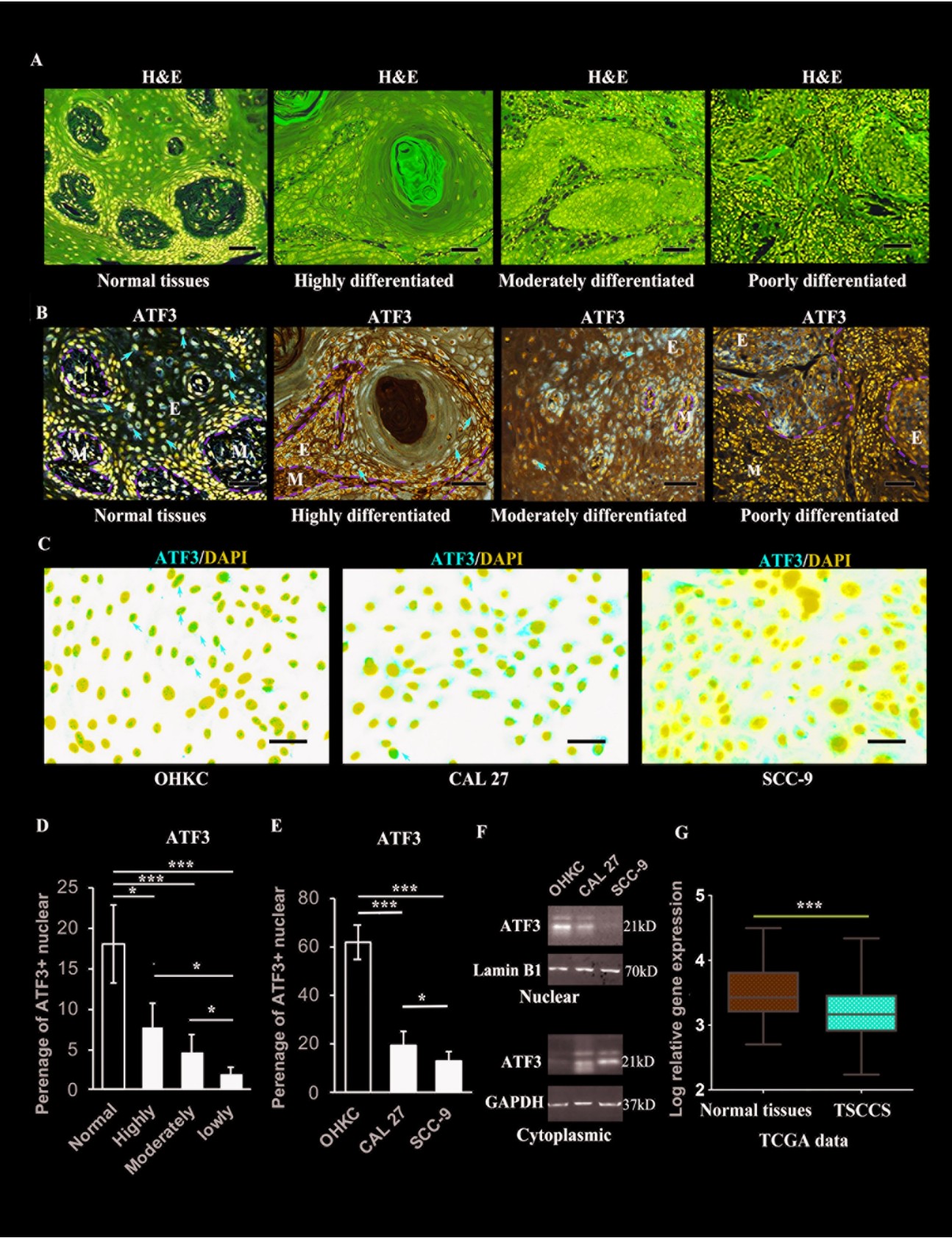

**Fig 1. ATF3 negatively correlates with the progression of TSCCs. A.** Representative images of histological analysis (H&E stain) of highly, moderately and poorly differentiated clinical TSCC tumors and normal oral tissues. **B.** Representative images of immunochemical staining of ATF3 from one patient of each different grade of TSCC tissues from **A,** red arrows indicate nuclear staining of ATF3, the green dot lines indicate the border between epidermal (E) and mesenchymal (M) compartments in the normal oral or tumor tissues. Lower magnification images together with representative images of ATF3 staining of two other patients from each different grade of TSCCs are shown in S1 Fig. **C.** Representative merged images of immunofluorescence staining of ATF3 (red) and DAPI (blue) in cultured primary oral human keratinocytes (OHKCs), CAL 27 and SCC-9 cells, DAPI for nuclei staining. Red arrows indicate nuclear staining of ATF3. Separate images of ATF3 and DAPI staining are shown in S3 Fig. **A-C:** Scale bars = 100 μm. **D.** Quantification of the percentage of normal oral epithelial cells and tumor cells with nuclear ATF3 staining (red arrows in **B**) calculated by counting the number of ATF3 nuclear stained cells in a total of 200 epithelial or tumor cells in **B**. **E.** Quantification of the percentage of nuclear positive staining of ATF3 in **C**. **F.** Nuclear and cytoplasmic fractions prepared from the cells in **C** were used for western blot analysis of ATF3. Lamin B1 was used as a loading control for nuclear fractions, GAPDH was used as a loading control for cytoplasmic fractions. **G.** Normalized expression level of ATF3 (relative log expression) in normal tissues and in oral tumor tissues derived from ATGC data. **D, E, F**: $^{*}p < 0.05$; $^{**}p < 0.01$; $^{***}p < 0.005$ when comparing two groups as indicated.

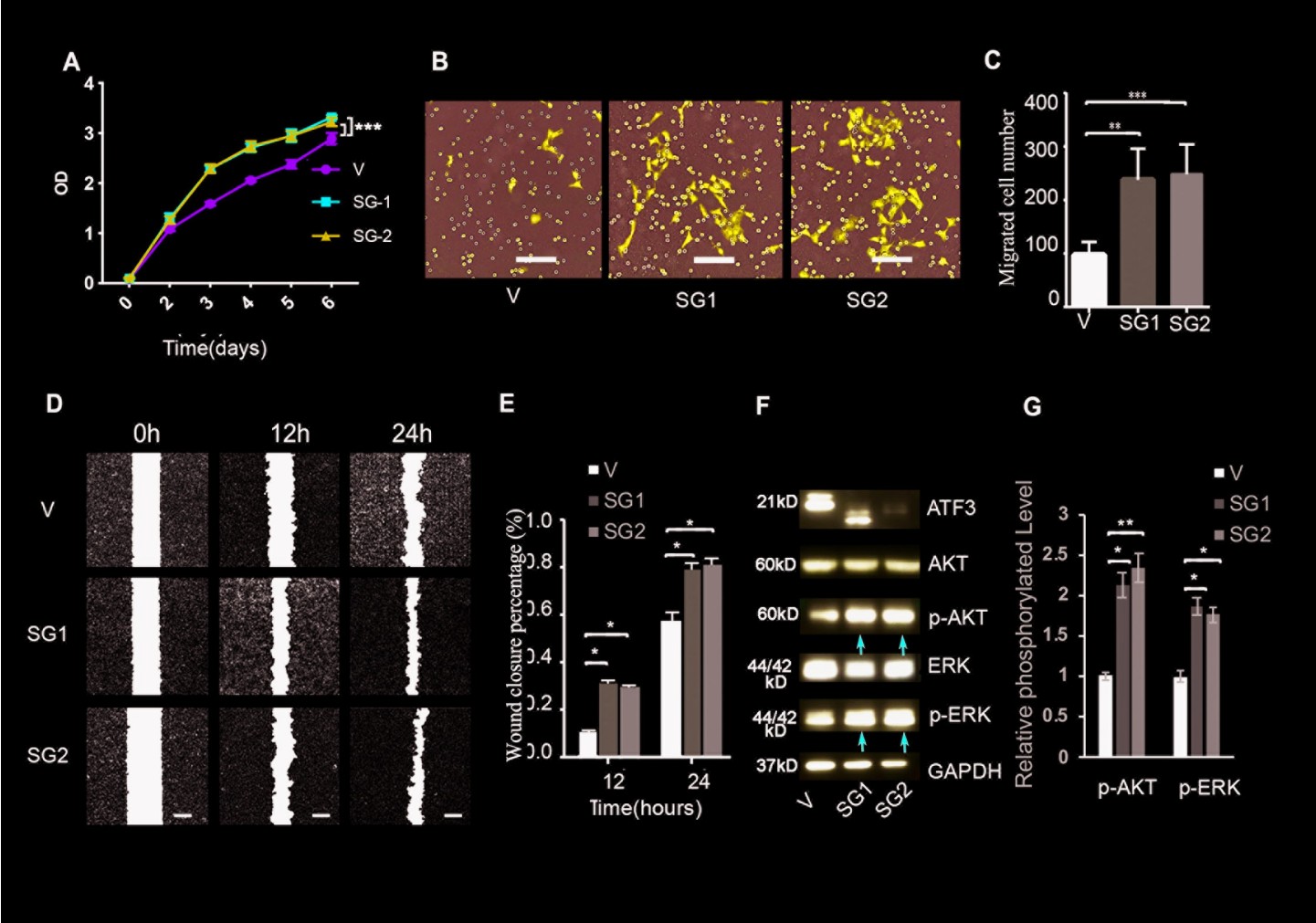

**Fig 2. Deletion of ATF3 enhances the growth and migration of TSCC cells. A.** Growth of CAL 27 cells with the CRISPR/Cas9 mediated deletion of ATF3 (SG1 or SG2) or with the empty vector as a control (V) were analyzed using a CCK8 kit at different time points. $^{**}p < 0.01$ when compared with the control group. **B, C.** Trans-well migration assays were performed with ATF3-deleted (SG1 or SG2) or control (V) TSCC cells; images of migrated cells at 24 h are shown in **B**, and the numbers of migrated cells in the different groups are shown in **C**. $^{**}p < 0.01$, $^{***}p < 0.005$ compared with the control group. Scale bars in **B** = 100 μm. **D, E.** Wound-healing assays were performed with ATF3-deleted (SG1 or SG2) or control (V) TSCC cells at the indicated times, representative wound healing images are shown in **D**, and percentage of wound closure is calculated in **E**. $^{**}$ $p < 0.01$ when compared with the control group. Scale bars in **D** = 200 μm. **F.** Western blot analysis of the indicated protein levels in control and in ATF3-deleted TSCCs. Red arrows indicate increased band density in ATF3-deleted cells compared with control cells (V). **G.** Quantification of phosphorylated AKT (p-AKT) and ERK (p-ERK) levels normalized with the total level of corresponding proteins. $^{*}$ $p < 0.05$, $^{**}$ $p < 0.01$ when compared with the control group as indicated.

relatively high expression of ATF3 (S2 and S7 Figs). The inhibition of TSCC growth and migration by disruption of ATF3 expression was found to be correlated with the increased activation of AKT and ERK signaling pathways, as shown by significant increases of pAKT and pErk, while total levels of those factors were unaffected (red arrows, Fig 2F and 2G) and the efficient deletion of ATF3 was also confirmed (Fig 2F). These data indicate that the loss of ATF3 function enhances TSCC cell growth and migration.

## High expression levels of ATF3 suppress TSCC cell growth and migration

Next, we tested the effects of increasing ATF3 expression in oral SCC cell lines. We stably over-expressed ATF3 in SCC-9 cells, which showed relatively low expression levels of ATF3 (S2 Fig). The high expression level of ATF3 in SCC-9 cells infected with a retrovirus expressing ATF3 was confirmed (S5B Fig). Ectopic expression of ATF3 significantly suppressed the growth of SCC-9 cells (Fig 3A), and also inhibited their migration in Transwell assays (Fig 3B and 3C) and in wound healing migration assays (Fig 3D and 3E). These results were also validated using another TSCC cell line SCC-4 that overexpresses ATF3 (S8 Fig). The corresponding activation of AKT and ERK was also suppressed in SCC-9 cells overexpressing ATF3 (Fig 3F and 3G).

Taken together, our findings suggest that ATF3 negatively regulates the growth and migration of TSCC cells, possibly by downmodulating the activation of the AKT and ERK pathways.

## ATF3 downregulates its downstream targets IFI6 and IFI27, which are highly expressed in TSCC cells

In order to understand the molecular mechanism by which ATF3 negatively regulates the growth and migration of TSCC cells, we analyzed the mRNA gene expression profile of CAL 27 cells with the deletion of ATF3 using RNA-seq. Gene Ontology (GO) enrichment analysis was performed and the Top 15 upregulated pathways were identified in the ATF3 deleted (SG1) group (Fig 4A). The GO analysis showed that most of those pathways are cytokine and chemokine regulated pathways, including "regulation of immune cell migration", "response to virus/bacterium" and "response to type I interferon or interferon gamma", as well as "epidermal cell differentiation/development" (Fig 4A). Gene cluster analysis revealed the upregulation of cytokine genes including known ATF3 target genes, such as IL-6 and IL-1β, and agreed with the GO analysis (Fig 4B). Two members of the IFI (interferon (IFN) alpha-inducible) gene family, IFI6 and IFI27, were identified as the Top upregulated genes in the ATF3-deleted cells (Fig 4B). Since interferons, especially the cytokine interferon alpha (IFNα), have been shown to mediate diverse immune responses to tumors [26], we further investigated the roles of IFI6 and IFI27 in ATF3 deleted and in ATF3 overexpressing TSCC cells.

The upregulation of IFI6 and IFI27 was validated by RT-PCR analysis both in SG1 and in SG2 ATF3 deleted cells (Fig 4C). Conversely, the lower expression of both IFI6 and IFI27 was observed in SCC-9 cells that overexpress ATF3 compared to control cells (Fig 4D). To investigate whether IFI6 and/or IFI27 are direct targets of ATF3, we analyzed published ATF3 CHIP-seq data (http://cistrome.org) and found that both IFI6 and IFI27 are putative targets of ATF3 in several cell types, including colon cancer cells, liver epithelial cells, erythroblasts, embryonic stem cells and skin dermal fibroblasts [27–30]. To determine whether ATF3 directly targets IFI6 and/or IFI27 in TSCC cells, CHIP assays were performed. Those results confirmed that ATF3 can directly bind predicted ATF3 binding sites in the IFI6 and IFI27 promoter regions (Figs 4E, 4F and S9). That binding could be reduced by ATF3 deletion or enhanced by ATF3 overexpression (Fig 4E and 4F), but was not detected by control non-immune IgGs (S9 Fig). Taken together, these results suggest that ATF3 directly targets IFI6 and IFI27 to negatively control their expression in TSCCs.

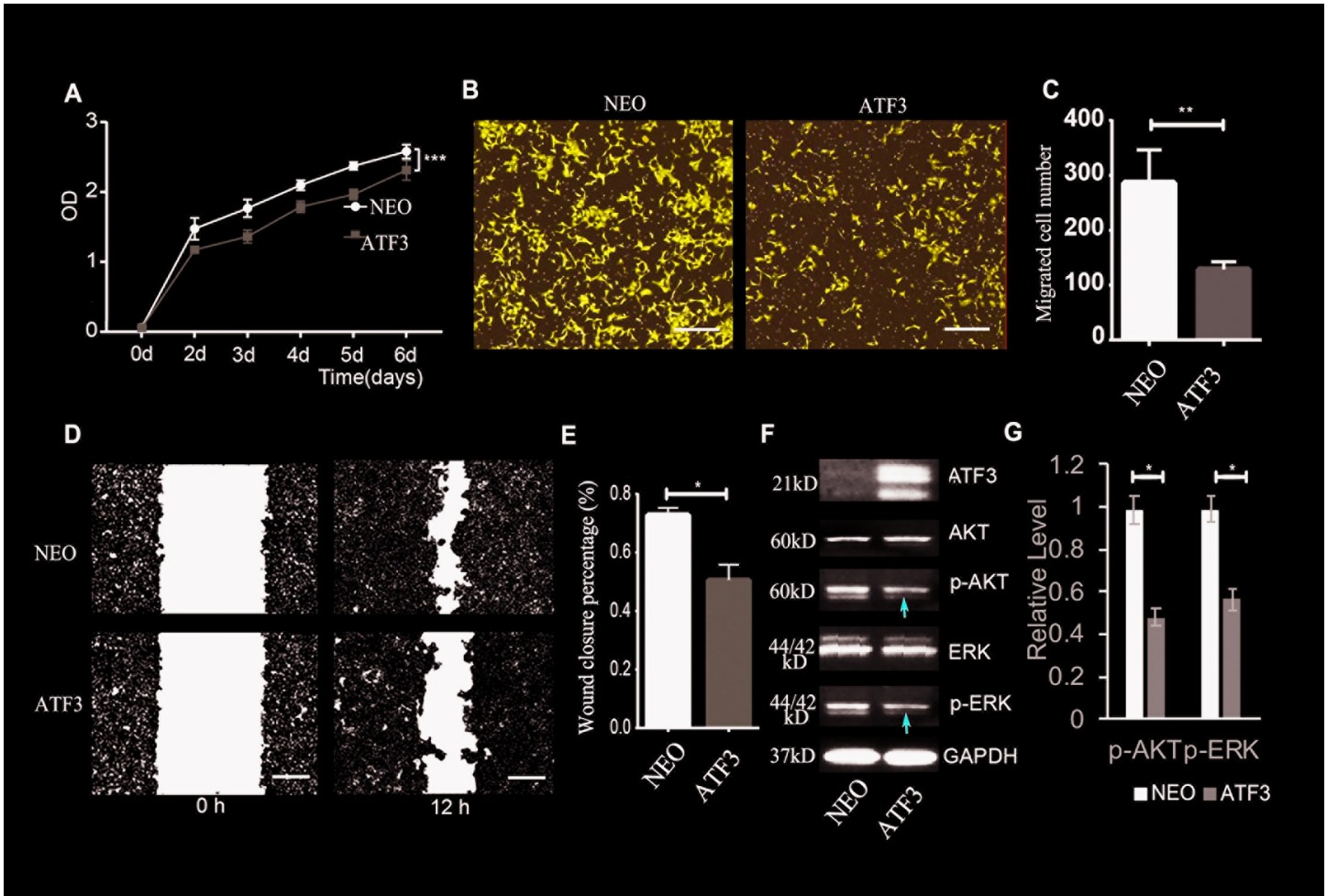

**Fig 3. Overexpression of ATF3 suppresses the growth and migration of TSCC cells. A.** Growth of SCC-9 TSCC cells infected with a retrovirus overexpressing ATF3 (ATF3) or expressing neomycin as a control (NEO) were analyzed using a CCK8 kit at different time points. ***p<0.005 compared with the control group. **B, C.** Transwell migration assays were performed with TSCC cells overexpressing ATF3 (ATF3) or control (NEO); images of migrated cells at 24 h are shown in **B**, and numbers of migrated cells through the filter in the different groups are shown in **C**. **p<0.01 compared with the control. Scale bars in **B** = 200 μm. **D, E.** Wound-healing assays were performed with TSCC cells overexpressing ATF3 (ATF3) or control (NEO) at the indicated times; representative images are shown in **D**, and percentage of wound closure is calculated in **E**. *p<0.05 compared with the control group. Scale bars = 200 μm. **F.** Western blot analysis of indicated protein levels in control NEO and in ATF3 overexpressing (ATF3) TSCCs. Red arrows indicate decreased band densities compared with the control group. **G**. Quantification of relative level (density) of the phosphorylated forms of AKT (p-AKT) and ERK (p-ERK) normalized with the total level of the corresponding proteins. *p<0.05 compared with ATF3 overexpressed cells with the control group as indicated.

IFI27 has already been reported to be highly expressed in oral SCCs [31]. We checked the expression level of IFI6 in TSCCs using IHC, and found high expression levels of IFI6 in clinical tumor tissues as well (Fig 4G and 4H). Importantly, TCGA data analysis further revealed that both IFI6 and IFI27 are highly expressed in TSCCs compared to normal tissues (Fig 4J). These data suggest that IFI6 and IFI27, which are direct targets of ATF3 and are negatively regulated by ATF3, are upregulated in TSCCs.

## ATF3 suppression of TSCC growth and migration *in vitro* depends on the expression of IFI6 and IFI27

Next, we tested whether ATF3 modulates the growth and/or migration of TSCC cells through the regulation of IFI6 and/or IFI27. First, we confirmed the knockdown efficiency of IFI6 and

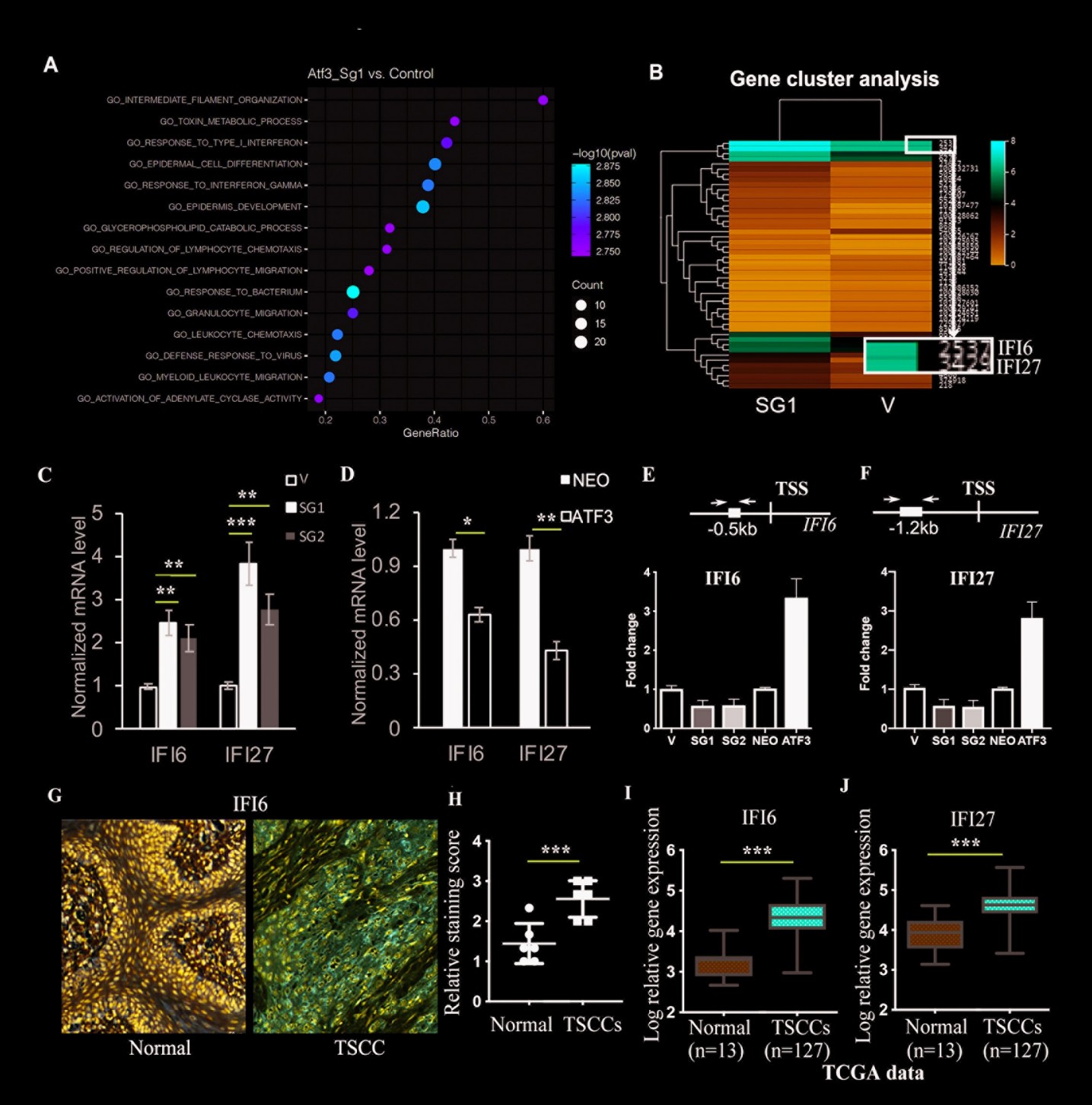

**Fig 4. ATF3 suppresses IFI6 and IGI27, which are highly expressed in TSCCs. A.** RNA-seq analysis of gene expression profiles in ATF3-deleted (SG1) and in control (V) CAL 27 TSCC cells. Dot plot of enriched GO terms. The 15 GO processes with the largest gene ratios are shown in order of gene ratio. Dot size represents the number of genes with increased expression in SG1 cells compared to control cells (V) associated with that GO term. Dot color indicates the log p-value. **B.** IFI6 and IFI27 were identified in the most upregulated genes in the SG1 versus the control V cell with gene cluster analysis. **C.** RT-PCR analysis of IFI6 and IFI27 expression in ATF3-deleted (SG1 and SG2) and in control (V) TSCC cells. **D.** RT-PCR analysis of IFI6 and IFI27 expression in ATF3-overexpressed (ATF3) and in control (NEO) TSCC cells. **E, F:** Extracts of CAL 27 cells with either deletion of ATF3 (SG1 or SG2) or overexpression of ATF3 were processed for chromatin immunoprecipitation (CHIP) assays with anti-ATF3 antibodies followed by RT-PCR analysis of IFI6 (**E**) or IFI27 (**F**) promoter regions containing an ATF3 binding site (black box in the maps above). TSS: Transcriptional Start Site. The CHIP products were also analyzed by normal PCR, and images of PCR products by gel electrophoresis are shown in S9 Fig. **G.** IHC analysis

of IFI6 expression in 6 clinical TSCC tumors and in 6 normal tissues. **H**. Quantification of IFI6 staining (n = 6) in G. **I**. IFI6 expression in TSCC tissues and in normal tissues from the ATGC database. **J**. IFI27 expression in TSCC tissues and in normal tissues from the ATGC database. Both **I** and **J** are shown with relative log expression for the Y-axis. **C, D, H-J**: * p<0.05, ** p<0.01 and *** p<0.005 when compared with the corresponding control group as indicated.

IFI27 by transfection of three independent siRNAs (S9A and S9B Fig), then we knocked down IFI6 or IFI27 in ATF3-deleted CAL 27 cells (sg1+siIFI6 or sg1+siIFI27). The results showed that the knockdown of IFI6 or IFI27 significantly blocked the increased growth of CAL 27 cells induced by the deletion of ATF (Fig 5A), and the knockdown efficiency of IFI6 and IFI27 in ATF3-deleted cells was also validated (Fig 5B). Next, we used a lentivirus expression vector to overexpress IFI6 and IFI27 in SCC-9 cells, then infected those cells with a retrovirus expressing ATF3. We found that the overexpression of IFI6 or IFI27 together with ATF3 (ATF3+IFI6 or ATF3+IFI27) counteracted the reduced growth and migration of SCC-9 cells with only the overexpression of ATF3 (ATF3+GFP) (Fig 5C), and the high expression levels of IFI6 or IGI27 were also verified by qRT-PCR (Fig 5D).

To analyze whether ATF3 regulates TSCC migration depending on the expression of IFI6 or IFI27, *in vitro* wound healing assays were carried out with ATF3 deleted cells with either knockdown IFI6 or IFI27 (SG1+siIFI6 or SG1+siIFI27) (the knockdown efficiencies of IFI6 or IFI27 are shown in S10 Fig) or the overexpression of IFI6 or IFI27 in ATF highly expressing cells (ATF3+IFI6 or ATF3+IFI27). We found that knockdown of either IFI6 or IFI27 inhibited the migration increased by the deletion of ATF3 (Figs 5E and S11A) and that the forced expression of IFI6 or IFI27 could rescue the migration inhibition induced by the high expression of ATF3 (Figs 5F and S11B). The enhancement of TSCC migration by IFI6 and by IFI27 was further confirmed using trans-well migration assays (Figs 5G and S11C). Importantly, the corresponding changes of AKT or ERK activity induced by the deletion or overexpression of ATF3 were counteracted by IFI6 or IFI27 expression (Fig 5H–5K).

These data suggest that ATF3 modulates the growth and migration of TSCC cells through the downregulation of IFI6 and IFI27.

## ATF3 negatively modulates TSCC growth through the regulation of IFI6 and IFI27 *in vivo*

To validate the biological role of ATF3 in TSCCs through IFI6 and IFI27 *in vivo*, we first tested whether the inhibition of ATF3 promotes tumor growth of TSCCs in mice. CAL 27 cells with or without the CRISPR-Cas9 mediated deletion of ATF3 were xenografted on the dorsal skin of nu/nu mice. One month after grafting, the mice were sacrificed and the grafted tumors were collected (S12A Fig). ATF3-deficient TSCC cells (SG1 and SG2) had formed significantly larger tumors than control cells (V) (Fig 6A and 6B). The proliferation and differentiation status of the tumors formed were then evaluated by IHC. Ki-67 staining showed that there were many more Ki-67 positive cells (red arrows) in tumor tissues with the deletion of ATF3 (SG1 or SG2) (Figs 6C, 6D and S12B). All tumor cells from the different groups were positive for the epidermal basal cell marker keratin 14 (K14) (Figs 6C, 6D and S12C), which is normally expressed in the oral stratum basal layer (undifferentiated cells)[32]. IHC of keratin 4 (K4) and keratin (K13), which have been reported to be expressed in normal oral differentiating suprabasal cells [32], but not in poorly differentiated oral SCCs [33], was positive but neither of those markers was clearly expressed in TSCC cells from any of the groups (S12D–S12F Fig). These data indicate that the loss of ATF3 could significantly enhance TSCC growth *in vivo*.

In contrast, ATF3 overexpressing SCC-9 cells (ATF3+GFP) generated smaller tumors than SCC-9 cells transfected with an empty vector (NEO+GFP) (Figs 6E, 6F and S13A). The effects of ATF3 overexpression on tumor growth were diminished by the overexpression of IFI6 or

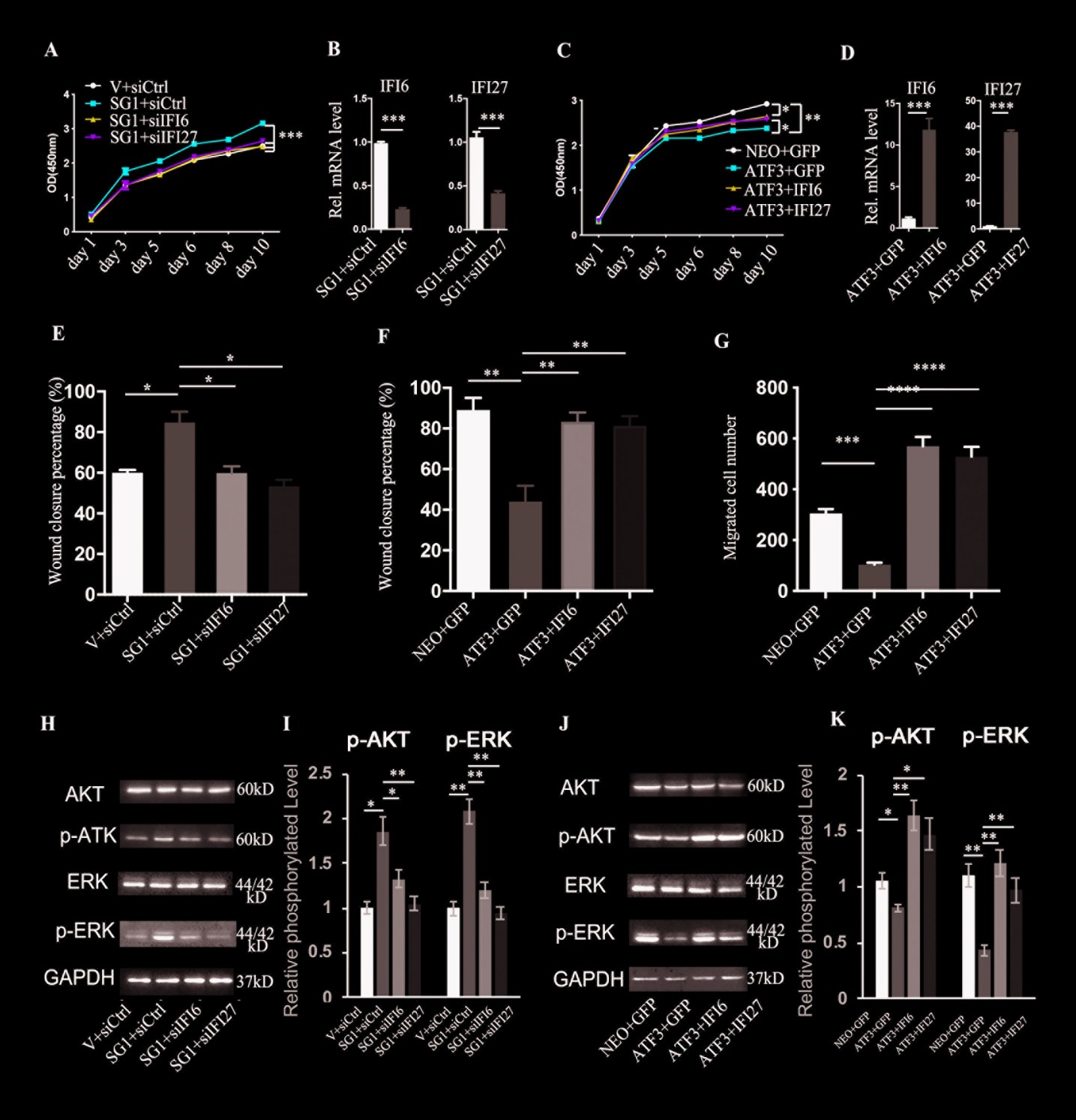

**Fig 5. ATF3 regulates TSCC cell growth and migration through the modulation of IFI6 and IFI27. A.** Growth of CAL 27 TSCC cells with the CRISPR/Cas9 mediated deletion of ATF3 (SG1) plus either knockdown of IFI6 (SG1+siIFI6) or IFI27 (SG1+siIFI27) or scramble siRNA (SG1+siCtrl) and an empty vector together with a scrambled siRNA as a control (V+siCtrl) were analyzed using a CCK8 kit at different time points. **B.** RT-PCR analysis of IFI6 or IFI27 from TSCC cells pre-infected with a CRISPR/Cas 9 ATF3 virus, then transfected with siRNAs of IFI6 or IFI27, at 48 h after transfection. **C.** Growth of SCC-9 cells infected with a retrovirus overexpressing ATF3 (ATF3) plus stable expression of IFI6 (ATF3+IFI6) or IFI27 (ATF3+IFI27) or GFP(ATF3+GFP) and with expressing neomycin and GFP as a control (NEO+GFP) were analyzed using a CCK8 kit at different time points. **D**. RT-PCR analysis of IFI6 or IFI27 from TSCC cells at 48 h after infection with ATF3 expressing virus together with either IFI6 or IFI27 expressing retrovirus as in **C. E,** Wound healing migration assays were performed with CAL 27 cells in the same groups as in **A,** the healing

percentage at 24 h after wounding is shown and images of wound healing are shown in S11A Fig. **F**. Wound healing migration assays were performed with SCC-9 cells in the same groups as in **C**, the healing percentage at 24 h after wounding is shown, and images of wound healing are shown in S11B Fig. **G**. Transwell migration assays were performed with cells in the same groups as in **C**. The numbers of migrated cells in the different groups are shown, and images of migrated cells at 24 h are shown in S11C Fig. **H**. Western blot analysis of the indicated protein levels in cells in the same groups as in **A**. **I**. Quantification of phosphorylated AKT (p-AKT) and ERK (p-ERK) levels normalized with the total levels of the corresponding proteins in **H**. **J**. Western blot analysis of the indicated protein levels in cells in the same groups as in **C**. **K**. Quantification of phosphorylated AKT (p-AKT) and ERK (p-ERK) levels normalized with the total levels of the corresponding proteins in **J**. * **A-K:** *p<0.05, **p<0.01, ***p<0.005, ****p<0.001 compared with the corresponding control groups as indicated.

IFI27 (ATF3+IFI6 or ATF3+IFI27), which were further supported by staining of the cell proliferation marker Ki-67 (red arrows), which showed that the overexpression of ATF3 significantly suppressed the proliferation of tumor cells, and that suppression was blocked by the stable expression of IFI6 or IFI27 (Figs 6G, 6H and S13B). All tumor cells from the different groups expressed K14, however, we found a lower expression of K14 in TSCC cells induced by the high expression of ATF3, but not in cells with co-expression of ATF3 and IFI6 or IFI27 (Figs 6G, 6H and S13C). Interestingly, we also observed that higher levels of K4 and K13 were expressed in ATF3 over-expressing cells (red arrows), but again not in cells co-expressing ATF3 and IFI6 or IFI27 (Figs 6K–6N and S13D and S13E). These data suggest that ATF3 can inhibit tumor cell growth and affect TSCC cell differentiation *in vivo*.

Taken together, these results suggest that ATF3 negatively modulates TSCC tumor growth and differentiation *in vivo* through the expression level of IFI6 or IFI27.

## Discussion

ATF3, a stress-response gene, is expressed at low levels in normal cells but can be induced by a variety of signals including hypoxia, anoxia, DNA damage and chemicals to play important roles in cellular processes that respond to signals disrupting homeostasis such as tissue damage [5]. The role of ATF3 in cancers has been extensively investigated and accumulated evidence has demonstrated that ATF3 can play tumor promoting or tumor suppressing functions depending on the type of tumor cell and the stromal context [10,12,13,15–18], and particularly, on the conditions of the cells as well [6]. For instance, the loss of ATF3 was found to promote prostate cancer development, but the overexpression of ATF3 enhanced the metastasis of prostate cancer [34,35]. ATF3 has been reported to play dichotomous roles by enhancing normal mammary epithelial cell apoptosis and by promoting breast cancer cell proliferation [8,36] and a similar situation was observed in TSCCs. In this study, IHC analysis revealed that ATF3 is localized in the nuclei of normal tongue epithelial cells and that nuclear expression levels of ATF3 are negatively correlated with the differentiation status of clinical TSCCs. This finding is inconsistent with two recent studies that showed higher expression levels of ATF3 in clinical TSCCs compared to normal tissues [23,24]. This difference in results is likely due to the conditions of the clinical tumors. Interestingly, the overexpression of ATF3 driven by the bovine keratin 5 (BK5) promoter in mice didn't result in the development of TSCCs, but did elicit the formation of oral mucosa SCCs [37], indicating a complicated role of ATF3 in oral SCCs. Studies by Shi et al. and by Hu et al. identified ATF3 as a downstream target of microRNAs that play important roles in TSCCs [23,24], but the direct effect of ATF3 on TSCCs has not been fully explored. Moreover, neither of those studies investigated the subcellular localization of ATF3 in TSCC cells. Both the nuclear and cytoplasmic localization of ATF3 has been reported in different situations related to the cell type [38], and the nuclear localization of ATF3 has been linked to its activation [39–41]. From our observations, we propose that ATF3 loses its nuclear localization with the progression of TSCCs. Importantly, analysis of the ATGC database showed that there was a significantly lower expression of ATF3 in tumor samples compared to normal tissues (P = 0.0037, Fig 1D), which supports our findings. Our gain- or loss-

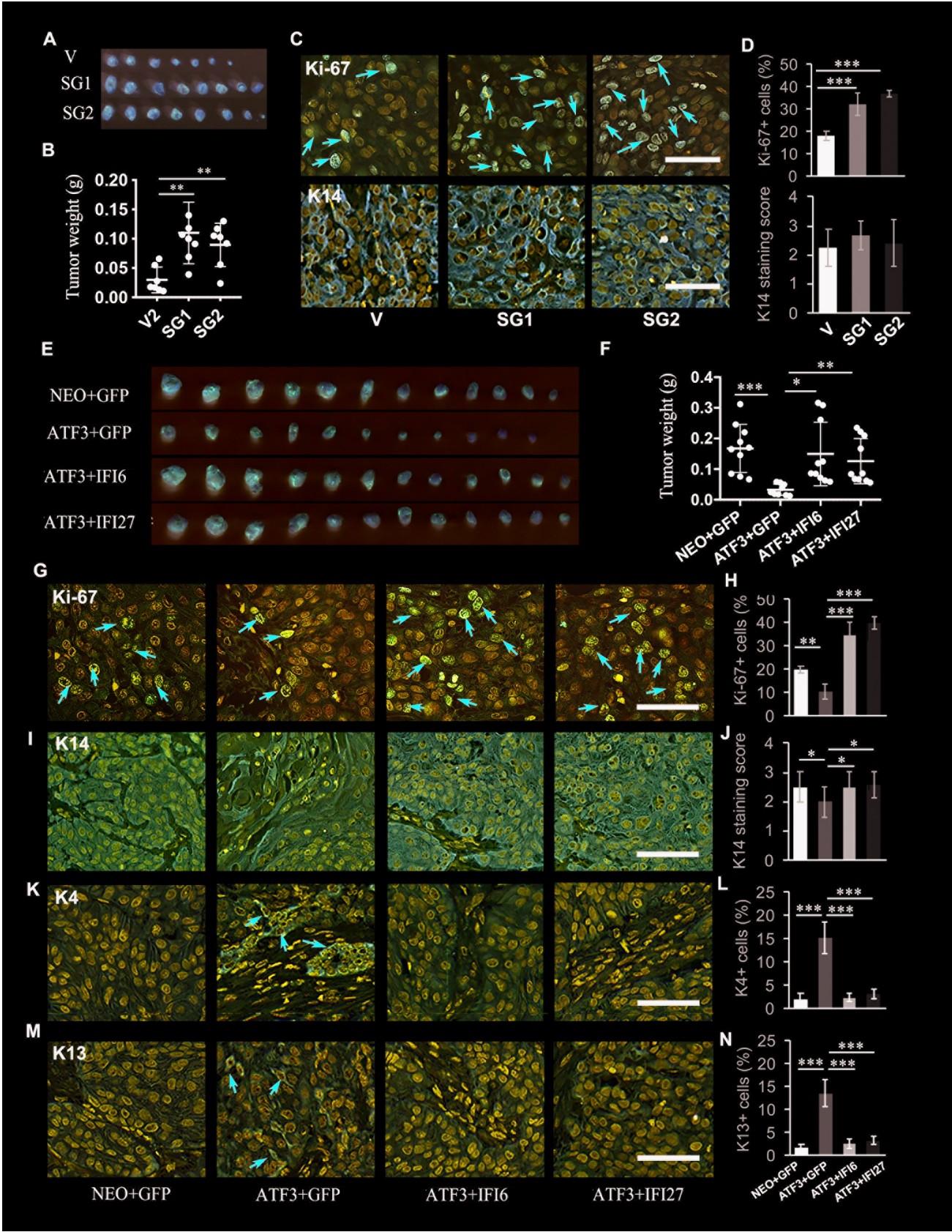

**Fig 6. ATF3 negatively regulates OSCC cell growth *in vivo* through the regulation of IFI6 and IFI27 expression. A.** Images of tumors harvested from grafts of CAL 27 cells with the deletion of ATF3 (SG1 or SG2) or the control group (V). Representative **i**mages of mice are shown in S12A Fig. **B.** The weights of tumors from (**A**) in each group were measured; each closed circle represents one tumor. **C.** Representative images of Ki-67 and Keratin (K14) staining of tumors from **A**. Red arrows indicate Ki-67 positive cells. Lower magnification of Ki-67 and K14 staining images are shown in S12B and S12C Fig. **D.** Quantification of Ki-67 positive cells and relative K14 staining scores in **C. E.** Images of tumors harvested from grafts of SCC-9 cells with overexpression of ATF3 (ATF3) plus overexpression of IFI6 (ATF3+IFI6) or IFI27 (ATF3+IFI27), and NEO as a control group. Representative **i**mages of mice are shown in S13A Fig. **F.** The weights of tumors from (**E**) in each group were measured and are shown in the graph; each closed circle represents one tumor. **G,I,K,M.** Representative images of Ki-67 (**G**), Keratin 14 (K14, **I**), Keratin 4 (K4, **K**) and Keratin 13 (K13, **M**) staining of tumors from **E**. Red arrows indicate Ki-67 positive cells in **G**, K4 positive cells in **K** and K13 positive cells in **M**. Lower magnification of Ki-67, K14, K4 and K13 staining images are shown S13B–S13E Fig. **H, J, L, N.** Corresponding quantification of **G, I, K** and **M** as labeled. All quantification data were from 6 tumors of each group (n = 6): $^*$p<0.05, $^{**}$p<0.01, $^{***}$p<0.005 compared with the two corresponding control groups as indicated. All scale bars = 100 μm.

of function analyses of ATF3 in four different TSCC cell lines (Figs 2 and 3) showed that ATF3 negatively modulates the growth and migration of TSCC cells associated with activation of the AKT/ERK pathways. Furthermore, suppression of TSCC growth by ATF3 was verified in an *in vivo* mouse model, which also showed that ATF3 could promote TSCC cell differentiation (Fig 6K–6N), agreed to GO analysis result of RNA-seq data (Fig 4A) and the induction of differentiation has been considered to be an anti-tumor function of ATF3 [42]. Therefore, we conclude that ATF3 plays an anti-tumor function in human TSCCs.

Consistent with our study, the tumor suppressing function of ATF3 has been reported in multiple studies with different types of tumors. ATF3 was found to be downregulated in several human cancers, including liver cancer, colon cancer, bladder cancer, prostate cancer and other epithelial cancers [12,17,18,20–22]. It has been shown that ATF3 negatively regulates the progression of different cancer cells through the modulation of different downstream pathways. As an example, in skin keratinocytes, ATF3 targets p53 to suppress cellular senescence to promote keratinocyte tumorigenesis [13]. However, ATF3 suppresses esophageal SCC growth by downregulating ID1 (inhibitor of DNA binding 1), blocks the metastasis of bladder cancer by regulating gelsolin-mediated remodeling of the actin cytoskeleton and inhibits the progression of hepatocellular carcinoma cells by upregulating CYR61 expression. Recently we reported that ATF3 suppresses expression of the cytokine IL-6 in dermal fibroblasts to inhibit melanoma cell growth through paracrine pathways [43]. Taken together, these results suggest that ATF3 negatively regulates tumor progression probably through specific downstream targets. In our study, RNA-seq analysis revealed that "response to type I and type II interferon" pathways were upregulated in ATF3 deleted TSCCs, and two IFN inducible genes, IFI6 and IFI27, were upregulated in ATF3-deleted TSCCs, and were downregulated in ATF3-overexpressing cells. Both IFI6 and IFI27 are type I interferon alpha inducible proteins, which are encoded by genes with high expression levels in microarray analysis of OSCCs compared to normal tissues [44]. However, the detailed functions of IFI6 and IFI27 have not been fully characterized. Both IFI6 and IFI27 were highly expressed in our clinical TSCC tumors, and again, that finding was supported by analysis of the TCGA database. Importantly, our *in vitro* study showed that the overexpression of IFI6 or IFI27 could counteract the inhibition of OSCC cell growth and migration induced by ATF3 overexpression and vice-versa, the knockdown of IFI6 or IFI27 blocked increased cell growth and migration with corresponding activation of the ERK/AKT pathways induced by the deletion of ATF3. and this finding was further verified using an *in vivo* mouse model.

IFNs, a well-known family of cytokines, play an essential role in antiviral effects, and the function of IFNs modulate cellular proliferation and immune responses mainly through interferon stimulating genes (ISGs). Both IFI6 and IFI27 are type I IFNalpha ISGs, and increasing evidence shows that IFNalpha plays dual opposing roles in cancer development based on ISGs, which determine whether it has anti- or pro-tumorigenic functions. IFI27 has been reported

to be involved in different biological processes including the regulation of cellular apoptosis; modulation of the transcriptional activity of nuclear receptors NR4A1, NR4A2 and NR4A3; involvement in the ubiquitin-mediated proteasomal degradation pathway and playing antivirus functions[45–49], IFI27 has been shown to inhibit Heptatis C virus replication by through the activation of JAK/STAT signaling pathway[50], which has been reported to be interact with AKT and MAPK/ERK pathways and to be associated with cancer development[51,52]. IFI27 has been also shown to play a tumor promoting function in multiple cancers and the knockdown of IFI27 was reported to inhibit the proliferation and invasion of TSCC cells [31], but the underlying molecular mechanism by which IFI27 regulates tumor cell growth and migration is still not clear. IFI6, has been shown to play anti-apoptosis role in gastric cancer cells, breast cancer cells, human myeloma cells and vascular endothelia cells mainly through inhibiting activation of caspase 3 and suppressing proapoptotic gene BCL-2 expression[53–56], and anti-apoptosis plays a crucial role for tumor development. Especially, IFI6 was recently reported to promote metastatic potential of breast cancer cells through mitochondrial ROS (mtROS)[57]. Therefore we conclude that ATF3 plays an anti-tumor function in TSCCs likely through the negative regulation of expression of its downstream targets IFI6 and IFI27. Future studies will be aimed at characterizing whether IFI6 and IFI27 control the growth and migration of TSCCs through regulation of JAK/STAT, anti-apoptotic or other potential downstream signaling pathways.

## Materials and methods

### Cell lines and culture

All TSCC cell lines, including Cal 27, SCC-9, SCC-25 and SCC-4 were purchased from the ATCC (Manassas, VA, USA). SCCO22 was derived from our previous publication[58]. All cells were grown in DMEM (Thermo Fisher Scientific, Waltham, MA, USA, #11965118) medium containing 10% fetal bovine serum (Biological Industries, Kibbutz Beit Haemek, Israel, #1809249),100 IU/ml penicillin and 100 mg/ml streptomycin (Thermo Fisher Scientific, #1935444).

Oral human keratinocytes (OHKCs) were isolated from tongue tissues according to a previously published protocol for isolation of keratinocytes from skin tissues[59]. The OHKCs were cultured in keratinocyte serum free medium (K-SFM, Thermo Fisher Scientific, #10725018) in culture dishes pretreated with a coating matrix containing type-I collagen (Gibco, R-011-K) and the medium was changed every 2 days.

### Tumor tissues from patients

Clinical normal tongue and TSCC tissues were obtained from patients who had provided verbal and written informed consent at the Stomatological Hospital, Shandong University. The procedures for obtaining those tissues were approved by the Medical Ethical Committee of the School of Stomatology, Shandong University (Protocol No. 20150401, Date: 12-05-2015).

### Plasmids and expression vectors

To delete ATF3 using the CRISPR/Cas9 system, single guide RNAs (sgRNAs) targeting ATF3 were designed using the MIT CRISPR Design tool (http://cripr.mit.edu/)). Potential target sequences were ranked according to the likelihood of off-target cuts. Paired DNA oligonucleotides encoding the target sequences of sgRNAs (ATF3 sgRNA1: Oligo1 5'-CACCGT-GAGCCCGGACAATACACGT-3', Oligo2 3'-AAACACGTGTATTGTCCGGGCT CAC-5'; ATF3 sgRNA2: Oligo1 5'-CACCGTCAAACACCAGTGACCCAGG-3', Oligo2 3'-

AAACCCTGGGTCACTGGTGTTTGAC-5) were cloned into a lentivirus vector pSpCas9
(BB)-2A-Puro(PX459)V2.0 (Addgene, Watertown, MA, USA, Plasmid #62988), which
expresses SpCas9 under the transcriptional control of a cytomegalovirus (CMV) promoter
[60]. To stably express IFI6 or IFI27, the corresponding cDNAs were subcloned into a lentivi-
rus vector pLent-EF1a-FH-CMV-GFP-P2A-puro. The retrovirus vector used to overexpress
ATF3 was derived from our previous publication [13].

## Virus production and infection

Retrovirus and lentivirus preparations and infections were performed as previously described
[13]. Viral vectors with corresponding packaging vectors pCL-Eco (Addgene, Plasmid
#12371), for retrovirus, and pMD2.G (Addgene, Plasmid #12259) together with psPAX2
(Addgene, Plasmid #12260) for lentivirus, were transfected into 293T cells using Lipofecta-
mine 2000 (Invitrogen, Carlsbad, USA, #L3000015). Starting one day after transfection, the
virus-containing supernatants were collected daily for three days and were stored at -80˚C. For
virus infection, $3 \times 10^5$ TSCC cells were seeded into 10 cm dishes. The following day, 6 ml
virus-containing medium with 8 μg/ml polybrene were added to the culture medium. After 6
h incubation, the virus-containing medium was removed and changed to normal growth
medium. After 48 h of infection, the infected TSCC cells were selected by adding 1 μg/ml puro-
mycin into the growth medium for another 6 days. Surviving cells were analyzed by western
blot analysis to validate the deletion or overexpression efficiency of the corresponding genes.

## Cell proliferation assays

Cell proliferation assays were carried out using a Cell Counting Kit-8 (CCK-8; Dojindo,
Tokyo, Japan). Three $\times 10^3$ cells per well were plated into 96-well plates and were cultured as
indicated in the Figures. At 0, 2, 3, 4, 5 and 6 days, 10 μl CCK-8 working solution were added
to each well, and incubated for 1.5 h at 37˚C. Optical densities were then measured at a wave-
length of 450 nm using a microplate reader (Spectrostar Nano, BMG Labtech, Offenburg,
Germany).

## Trans-well migration assays

Cell migration assays were performed using 8.0 μm pore size Transwell TM permeable sup-
ports (Corning Costar, Lowell, MA). Three $\times 10^4$ TSCC cells were seeded in the upper chamber
of each insert, which was placed into 6-well plates with medium containing 5% FBS. At 24 h,
cells remaining on the upper filter were scraped off gently using a cotton swab and the inserts
were gently washed with PBS. Cells that had migrated to the lower chamber were fixed with
4% paraformaldehyde for 10 min and then stained with 0.5% crystal violet solution for 20 min.
Filters were then imaged in a microscope. Five non-overlapping, representative images were
captured for each insert and were used to manually count the number of cells. Results are pre-
sented as the mean number of cells per field ± standard deviation.

## Wound-healing assays

Wound-healing assays were used to evaluate the migration ability of TSCC cells under condi-
tions indicated in the Figures. Briefly, $8 \times 10^5$ cells per well were seeded in 6-well plates. When
the cells reached nearly 90% confluence, the cell layer was scratched with a sterile 200 μl pipette
tip, then washed twice with PBS and cultured with medium containing 1% FBS. At 0,12 and 24
h, photographic images of the plates were acquired using a microscope.

## Quantitative RT-PCR (qRT-PCR) assay

Total RNAs were extracted from cells using an RNA Extraction kit (Takara Bio Inc., Kusatsu, Shiga Prefecture, Japan) according to the protocol of the manufacturer. The RNAs were dissolved in nuclease-free water and the concentrations were measured using a NanoDrop spectrophotometer. One μg of each total RNA was reverse transcribed to complementary DNA (cDNA) using a Takara PrimeScriptTM RT kit (Takara Bio Inc.). PCR reactions were performed with Takara SYBRR Premix Ex TaqTM II (Takara Bio Inc.) using a LightCyclerR 480 II (Roche Diagnostics Ltd., Rotkreuz, Switzerland). One hundred ng cDNA and 250 nmol gene specific primers were used for amplification in 20 μl qRT-PCR reactions. The PCR reactions were carried out at 95˚C for 30 s followed by 40 repeated cycles of 95˚C for 5 s and 60˚C for 20 s and then were terminated with an elongation step for 15 s at 72˚C. Data were acquired and analyzed with a LightCycler 480 system. The PCR primers used are listed in S1 Table.

## Chromatin immunoprecipitation (ChIP) assays

Chromatin immunoprecipitation (ChIP) assays were performed as described in the EpiQuik™ Chromatin Immunoprecipitation Kit protocol. Briefly, 1 x 10[6] CAL 27 cells were cross-linked with 1% formaldehyde, sonicated, precleared and then incubated with 2–5 mg antibody per reaction (rabbit monoclonal anti-ATF3 antibody (1:100), Abcam, Cambridge, MA, USA, #ab207434). Immune complexes were washed with low and high salt buffers, and the DNA was extracted and precipitated. The enrichment of the DNA template was analyzed by conventional PCR and qRT-PCR, using primers specific for the IL-27 and IL-6 gene promoters. The primers used are listed in S2 Table. The conventional PCR products were analyzed by gel electrophoresis, and the qRT-PCR followed the above procedure.

## Western blot analysis

To prepare whole cell protein extracts, the cells were washed twice with ice-cold PBS and then lysed with RIPA buffer containing 1% phenylmethylsulphonyl fluoride (PMSF) for 30 min on ice, followed by centrifugation at 12,000 r.p.m. at 4˚C for 10 min. The supernatants were collected, and protein concentrations were quantified using a BCA protein quantitation kit (Solarbio, Beijing, China). Equal amounts of protein per lane were separated on 10% SDS-PAGE gels (Beyotime, Shanghai, China) and were then electrotransferred to polyvinylidene fluoride (PVDF) membranes (Invitrogen) following the manufacturer's protocol. After blocking with 5% bovine serum albumin, the membranes were incubated with the primary antibody overnight at 4˚C with gentle shaking. The next day, blots were washed 3 times for 10 min each with 1xTBST (20 mM Tris-HCl, 150 mM NaCl), containing 0.05% (v/v) Tween-20 and then incubated with the secondary antibody for 2 h at room temperature. Finally, detection was performed using enhanced chemiluminescence reagents (Millipore, Billerica, MA, USA, #1702303). The following primary and secondary antibodies were used: rabbit monoclonal anti-ATF3 antibody (1:1000, #ab207434) and rabbit monoclonal anti-Lamin B1 (1:200, #ab194109) purchased from Abcam (Cambridge, MA, USA), rabbit polyclonal anti-AKT (1:1000, #4691), rabbit polyclonal anti-phospho-AKT (Ser473) (1:1000, #4060), rabbit polyclonal anti-Erk1/2 (1:1000, #4695), rabbit polyclonal anti-phospho-Erk1/2 (Thr202/Tyr204) (1:1000 CST, #4370) and rabbit polyclonal anti-GAPDH (1:2000, #2118) purchased from Cell Signaling Technology (CST) (Danvers, MA, USA) and the secondary antibody HRP-conjugated goat anti-rabbit (1:2000, #ab6721) from Abcam.

## Histology and Immunofluorescence analysis

Formalin-fixed tumor issues were embedded in paraffin and 10 μm sections were prepared for histological analysis following standard protocols. IHC analyses were performed as follows: each section was deparaffinized in xylene and post-fixed in ice-cold acetone, incubated with blocking buffer (2% bovine serum albumin and 5% donkey serum in PBS with 0.1% Triton 100) at room temperature for 1 h, and then incubated with the primary antibody at 4˚C overnight. On the second day, after removing the primary antibody solution and washing with PBS, each specimen was incubated with the secondary antibody at room temperature. After 1 h incubation, the sections were washed with PBS and stained with DAB (ZSGB-BIO) for 5 min. The following primary and secondary antibodies were used: rabbit monoclonal anti-ATF3 antibody (1:400, Abcam, #ab207434), rabbit polyclonal anti-IFI6 (1:200, Abcam, #ab192314), rabbit polyclonal anti-Ki-67 (1:500, Abcam, #ab15580), rabbit monoclonal anti-CK14 (1:1000, K14, Abcam, #51054), rabbit monoclonal anti-CK4 (1:200, Abcam, #51599), rabbit monoclonal anti-CK13 (1:200, Abcam, #92511) and the secondary antibody HRP-conjugated goat anti-rabbit (1:1000, #ab6721), all from from Abcam. For quantification of ATF3 staining, we calculated the percentage of cells with ATF nuclear positive staining by counting a total of 200 cells for each sample. For quantification of K4 and K13 staining, we calculated the percentage of cells with K4 or K13 positive staining by counting a total of 200 cells for each sample. For quantification of IFI6 and K14 staining (relative staining scores): All staining was performed twice and was evaluated twice by two independent persons. Evaluations were based on arbitrary units as follows: 1: no or weak staining, 2: intermediate staining and 3: strong staining, and the mean value of the independent measurements was taken as the final score. The number of samples for quantification data is shown in the corresponding Figure Legends.

## siRNA transfection

Transfection of siRNAs into TSCC cells followed the standard protocol. Briefly, TSCC cells at 70% confluence were transfected with 20 nM various siRNAs using Lipofectamine 3000 (Invitrogen). Seventy-two h later, the cells were collected for analysis by RT-PCR to determine knockdown efficiency. siRNA oligos that target IFI6, IFI27 and 36beta4 (used as an internal control) or a negative control scramble siRNA (siCtrl) were purchased from Gene Pharma (Shanghai, China). Three independent siRNAs for each gene were tested and the oligo sequences of all siRNAs are shown in S3 Table.

## RNA sequencing (RNA-seq) and data analysis

RNA sequencing of control (V) and ATF3-knockout CAL 27 cells (SG1) were performed using a BGISEQ-500 sequencing system at BGI (Shenzhen, China). Gene expression levels were normalized and quantified using the RSEM tool [61]. Differentially expressed genes (DEGs) between the V and SG1 groups were screened using the PossionDis method [62].

## *In vivo* tumor formation

To test whether the loss of ATF3 enhances TSCC tumor growth *in vivo*, 8-week-old female nu/nu nude mice (Crl:NU-*Foxn1*[nu] Immunodeficient Outbred)(Beijing Vital River Laboratory Animal Technology Co., Ltd., Beijing, China) were randomly assigned to three groups: a control group (n = 6, Cal 27 cells infected with CRISPR empty vector lentivirus, V), a SG1 group (n = 6, Cal 27 cells infected with CRISPR-sgRNA1 lentivirus) and a SG2 group (n = 6, Cal 27 cells infected with CRISPR-sgRNA2 lentivirus). To test whether ATF3 affects TSCC tumor growth through the regulation of IFI6 and/or IFI27, nude/nude mice were randomly divided

into the following four groups: a control NEO group (n = 6, SCC-9 cells infected with neomycin expressing retrovirus and an empty vector lentivirus, NEO), an ATF3 group (n = 6, SCC-9 cells infected with ATF3 expressing retrovirus and an empty vector lentivirus), an ATF3+IFI6 group (n = 6, SCC-9 cells infected with ATF3 expressing retrovirus and an IFI6 expressing lentivirus) and an ATF3+IFI27 group (n = 6, SCC-9 cells infected with ATF3 expressing retrovirus and an IFI27 expressing lentivirus). One x $10^6$ TSCC cells were injected subcutaneously into the dorsal skin of mice as described previously [13] and each mouse received a total of 4 injections. Tumors were observed and measured every day, and one month later, the mice were euthanized and their tumors were collected and weighed.

All animal procedures were performed in accordance with National and International Animal Welfare Regulations and with approval from the Stomatology School of Shandong University Committee on Animal Care.

## Statistical analysis

Statistical analysis was performed using GraphPad Prism 7 and data are presented as means ± standard error of the mean (SEM). All experiments were performed at least three times. Student's t-test was used to compare two groups and one-way or two-way ANOVA analysis were used for comparison of more than 2 groups. A $p < 0.05$ is considered to be statistically significant, and p values are labeled in the Figures.

## Supporting information

**S1 Fig. Nuclear staining of ATF3 correlates negatively with the prognosis of clinical TSCCs.** Immunochemical (IHC) staining of ATF3 was performed with different grade clinical TSCC tissues from Fig 1A, and representative staining images of 3 patients (cases 1, 2 and 3) from each group as indicated are shown. * indicates artificial bubbles. A high magnification image of Case 1 from each group is shown in Fig 1B. Scale bars = 100 μm.
(JPG)

**S2 Fig. ATF3 expression in different TSCC cell lines. A, B**. ATF3 expression levels in different TSCC cell lines were analyzed by RT-PCR (**A**) and by western blot (**B**). *p<0.05 when comparing the two groups as indicated.
(JPG)

**S3 Fig. Lower nuclear expression of ATF3 in TSCC cell lines.** Representative images of immunofluorescence staining of ATF3 (red, left panels) and DAPI (blue, middle panels) in cultured primary oral human keratinocytes (OHKCs), CAL 27 and SCC-9 cells. Merged images of ATF3 and DAPI staining are shown in the right panels, which are also shown in Fig 1C. DAPI for nuclei staining. Scale bars = 100 μm.
(JPG)

**S4 Fig. Low expression of ATF3 in HNSCs from the TCGA database.** Data analysis in **A** was performed at the following link: http://ualcan.path.uab.edu/cgi-bin/TCGAExResultNew2.pl?genenam=ATF3&ctype=HNSC; Data analysis in **B** was performed at the following link: http://ualcan.path.uab.edu/cgi-bin/TCGAExResultNew2.pl?genenam=ATF3&ctype=HNSC
(JPG)

**S5 Fig. Expression of ATF3 is mediated by CRISP-cas9 and a retro-expressing vector. A.** CAL 27 cells infected with a lentivirus carrying CRISPR/Cas9 with ATF3 sgRNA1 (SG1) or sgRNA2 (SG2) or with an empty vector as a control (V). After selection with puromycin, the cells were collected for western blot analysis of ATF3 protein to determine the deletion

efficiency. GAPDH was used as a housekeeping gene for a loading control. **B.** SCC-9 cells were infected with a retrovirus expressing ATF3 (ATF3) or neomycin (NEO) as a control; 48 h after infection, the cells were collected for RT-PCR (left panel) and western blot analysis for ATF3 expression. Relative mRNA levels of ATF3 were normalized with the 36beta4 gene, and GAPDH was used as a loading control for the protein level. ***p<0.005 compared with the control group. (JPG)

**S6 Fig. Knockdown of ATF3 promotes the growth of CAL 27 cells. A.** CAL 27 cells were transfected with two independent siRNAs of ATF3 or with a scrambled siRNA (siCtrl). 72 h after transfection, the cells were collected for RT-PCR analysis for ATF3 expression. The relative mRNA levels of IFI6 and IFI27 were normalized with the 36beta4 gene, ***p<0.005 compared with the control group (siCtrl) as indicated. **B.** Growth of CAL 27 cells transfected with siRNAs of ATF3 or with a scrambled siRNA (siCtrl) were analyzed using a CCK8 kit at different time points. **p<0.01 compared with the control siCtrl group. (JPG)

**S7 Fig. Deletion of ATF3 enhances the growth and migration of SCC-25 cells. A.** Growth of SCC-25 cells with the CRISPR/Cas9 mediated deletion of ATF3 (SG1 or SG2) or with the empty vector as a control (V) were analyzed using a CCK8 kit at different time points. **p<0.01 when compared with the control group. **B, C.** Trans-well migration assays were performed with ATF3-deleted (SG1 or SG2) or control (V) TSCC cells; images of migrated cells at 24 h are shown in **B**, and the numbers of migrated cells in the different groups are shown in **C**. **p<0.01 compared with the control group. Scale bars in **B** = 100 μm. **D, E.** Wound-healing assays were performed with ATF3-deleted (SG1 or SG2) or control (V) TSCC cells, representative wound healing images at 0 h and 24 h after wounding are shown in **D**, and the percentage of wound closure is calculated in **E**. ** p<0.01 when compared with the control group. Scale bars in **D** = 100 μm. (JPG)

**S8 Fig. Overexpression of ATF3 suppresses the growth and migration of SCC-4 cells. A.** Growth of SCC-4 TSCC cells infected with a retrovirus overexpressing ATF3 (ATF3) or expressing neomycin as a control (NEO) were analyzed using a CCK8 kit at different time points. **p<0.01 compared with the control group. **B, C.** Trans-well migration assays were performed with TSCC cells overexpressing ATF3 (ATF3) or control NEO; images of migrated cells at 24 h are shown in **B**, and the numbers of cells that migrated through the filter in the different groups are shown in **C**. **p<0.01 compared with the control. **D, E.** Wound-healing assays were performed with TSCC cells overexpressing ATF3 (ATF3) or control NEO (NEO); representative images at 0 h and 48 h after wounding are shown in **D**, and the percentage of wound closure at 48 h after wounding is calculated in **E**. *p<0.05 compared with the control group as indicated. Scale bars in **B** and **D** = 200 μm. (JPG)

**S9 Fig. ATF3 directly binds the promoter regions of the IFI6 and IFI27 genes. A,B.** Extracts of CAL 27 cells were processed for CHIP assays with the anti-ATF3 antibody and non-immune IgG followed by PCR analysis of either the IFI6 (**A**) or the IFI27 promoter regions containing ATF3 binding sites as shown (maps) in Fig 4E and 4F. Gel electrophoresis (1% agarose gel) shows the PCR amplified 160 bp fragment for IFI6 and the 463 bp fragment for IFI27. Labels for lanes: M: molecular markers, 1: Input, 2,3: ATF3 antibody; 4: IgG. **C,D.** Extracts of CAL 27 cells with either the deletion of ATF3 (SG1 or SG2) or the overexpression of ATF3 (ATF3) and the corresponding controls (V or NEO) were processed for CHIP assays with the anti-ATF3 antibody or non-immune IgG followed by PCR analysis of either IFI6 (**C**) or IFI27

(**D**) promoter regions containing ATF3 binding sites. PCR products were analyzed by gel electrophoresis. Labels for lanes: M: molecular markers, 1: Input, 2: V+ATF3 antibody, 3: SG1 +ATF3 antibody, 4: SG2+ATF3 antibody; 5: IgG, 6: NEO+ATF3 antibody, 7–8: overexpression of ATF3+ATF3 antibody.
(JPG)

**S10 Fig. siRNA-mediated knockdown of IFI6 or IFI27 expression in TSCC cells. A, B.** CAL 27 cells were transfected with 3 independent siRNAs of IFI6 (A) or IFI27 (B) or a scrambled siRNA (siCtrl). 72 h after transfection, the cells were collected for RT-PCR analysis of IFI6 and IFI27 expression. The relative mRNA levels of IFI6 or IFI27 were normalized with the 36beta4 gene, **p<0.01, ***p<0.005 compared with the control group (siCtrl) as indicated.
(JPG)

**S11 Fig. ATF3 controls TSCC migration through the negative regulation of IFI6 and IFI27. A,** Wound healing migration assays were performed with CAL 27 cells in the same groups as in Fig 5A. Images of wound healing at 24 h after wounding are shown here and quantification of the wound healing percentage is shown in Fig 5E. **B.** Wound healing migration assays were performed with SCC-9 cells in the same groups as in Fig 5C. Images of wound healing at 24 h after wounding are shown here and quantification of the wound healing percentage is shown in Fig 5F. **C.** Trans-well migration assays were performed with cells in the same groups as in Fig 5C. Representative images of migrated cells at 24 h after wounding are shown here, and the numbers of migrated cells in the different groups are shown in Fig 5G. Scale bars = 200 μm.
(JPG)

**S12 Fig. Loss of ATF3 promotes TSCC tumor growth *in vivo*. A.** Representative **i**mages of mice with TSCC tumors formed from grafts of CAL 27 cells with the deletion of ATF3 (SG1 or SG2) or the control group (V). **B-E.** Representative images of IHC staining for Ki-67 (**B**), K14 (**C**), K4 (**D**) and K13 (**E**) in tumors from the indicated groups as shown in **A**. High magnification images of Ki-67 and K14 staining are also shown in Fig 6C. Scale bars = 100 μm.
(JPG)

**S13 Fig. ATF3 not only suppresses TSCC growth but also promotes TSCC differentiation through the regulation of IFI6 and IFI27 *in vivo*. A.** Representative images of mice with tumors formed from grafts of SCC-9 cells with overexpression of ATF3 (ATF3) plus overexpression of IFI6 (ATF3+IFI6) or IFI27 (ATF3+IFI27), and NEO as a control group. **B-E.** Representative images of IHC staining of Ki-67 (**B**), K14 (**C**), K4 (**D**) and K13 (**E**) in tumors from the indicated groups as shown in **A**. High magnification images of these staining patterns are also shown in Fig 6G, 6I, 6K and 6M. Scale bars = 100 μm.
(JPG)

**S1 Table. Oligo sequences used for RT-PCR analysis**
(PDF)

**S2 Table. Oligo sequences used for CHIP assays**
(PDF)

**S3 Table. Oligo sequences used for siRNAs**
(PDF)

## Acknowledgments

We thank Ms. Xue Leng and other members of Wu's lab for helpful discussion and technical assistance.

## Author Contributions

**Conceptualization:** Cord Brakebusch, Xuxia Wang, Xunwei Wu.

**Data curation:** Lin Xu, Tingjian Zu, Tao Li, Min Li, Fuxiang Bai, Guanyi Liu, Jie Wen, Hui Li.

**Formal analysis:** Lin Xu, Xunwei Wu.

**Funding acquisition:** Xunwei Wu.

**Investigation:** Lin Xu, Tingjian Zu, Jun Mi, Xunwei Wu.

**Methodology:** Lin Xu, Tao Li, Fuxiang Bai, Hui Li, Xunwei Wu.

**Project administration:** Xunwei Wu.

**Resources:** Lin Xu, Min Li, Guanyi Liu, Xunwei Wu.

**Software:** Lin Xu, Jun Mi.

**Supervision:** Xuxia Wang, Xunwei Wu.

**Validation:** Xuxia Wang, Xunwei Wu.

**Visualization:** Lin Xu, Xunwei Wu.

**Writing – original draft:** Lin Xu, Xunwei Wu.

**Writing – review & editing:** Lin Xu, Cord Brakebusch, Xunwei Wu.

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
