## [Decision Letter · Decision Letter 0]

24 Jun 2020

Dear Dr Wu,

Thank you very much for submitting your Research Article entitled 'ATF3 downmodulates its new targets IFI6 and IFI27 to suppress the growth and migration of tongue squamous cell carcinoma cells' to PLOS Genetics. 

The manuscript was fully evaluated at the editorial level and by four independent peer reviewers. The reviewers appreciated the attention to an important problem, but raised some substantial concerns about the current manuscript. Based on the reviews, we will not be able to accept this version of the manuscript, but we would be willing to review again a much-revised version. We cannot, of course, promise publication at that time.

If you decide to revise the manuscript for further consideration at PLOS Genetics, please aim to resubmit within the next 60 days, unless it will take extra time to address the concerns of the reviewers, in which case we would appreciate an expected resubmission date by email to plosgenetics@plos.org.

[LINK]

We are sorry that we cannot be more positive about your manuscript at this stage. Please do not hesitate to contact us if you have any concerns or questions.

Yours sincerely,

Gregory S. Barsh

Editor-in-Chief

PLOS Genetics

Peter McKinnon

Section Editor: Cancer Genetics

PLOS Genetics

Reviewer's Responses to Questions

**Comments to the Authors:**

Reviewer #1: The review is uploaded as an attachment.

Reviewer #2: The manuscript reports that ATF3 expression and nuclear localisation negatively correlates with TSSC malignancy in vivo and growth and invasion in vitro. The low levels of ATF3 lead to upregulation of IFI6 and IFI27, which execute growth and migration of TSCCs. The manuscript reports the first association of ATF3 with IFI6 and IFI27. However, the study is preliminary and requires more work to become a strong paper. The ATF3 levels in cytoplasms and nuclei in tumor tissues and cultured TSCCs were not biochemically tested (fractionation), the growth differences of the cultured TSCCs with and without ATF3 is marginal, the Crispr-mediated deletion of ATF3 is not confirmed by sequence analyses, etc., etc. Moreover, it is unclear how ATF3 down regulates the IFI6/27, whether the IFI6/27 is dominant over the induction by the interferon produced upon white blood cell infiltration into the tumor tissues, and how IFI6/27 promotes growth, migration and invasion of TSCCs.

Reviewer #3: The work by Wang et al investigates the mechanisms of transformation in head and neck squamous cell carcinomas, specifically arising in the tongue epithelia (TSCC). Nuclear ATF3 is lower in less differentiated TSSC samples and in a public database. ATF3 represses the growth and migration of two TSCC cell lines and comprehensive gene expression analysis in one of these cell line plus/minus ATF3 gene deletion identified two targets IFI6 and IFI27. Both are highly expressed in TSCCs and their silencing blunts the increased growth/proliferation induced by ATF3 deletion. According to an epistatic role for IFI6 and IFI27, their overexpression counteracted the inhibitory effects of ATF3 overexpression.

The controversial positive/negative role of ATF3 in SCC has been previously reported, thus the novelty for the field is limited and the manuscript, in its present form, does not provide new insights into the mechanism(s) upstream or downstream of this stress-induced transcriptional regulator. It is not clear if IFI6 and IFI27 are direct targets for ATF3 and, more importantly, how IFI6 and IFI27 exert their effect(s). How do the present findings relate to a pro-tumorigenic function of ATF3 in early steps of SCC development as previously reported by Wu et al. (Nature 2010)? There is also the need to resolve some technical concerns regarding ATF3 nuclear localization in IHC, comparing normal oral KC vs TSSC and ChIP analyses of IFI6 and IFI27 promoters (with and without ATF3 silencing). Addressing these points would strengthen the enthusiasm for the manuscript.

Specific points

• A western blot of nuclear vs cytosolic fractions in SCC cells vs oral keratinocytes is required to hypothesize a cytosolic relocalization of ATF3 during transformation.

• The DAPI nuclear signal in the immunofluorescence in figure 1C should be separated to permit the detection of ATF3 nuclear signal. A confocal image of a higher magnification is required. On this line, a z section should be shown as the "nuclear" ATF3 red signal might be the result of a stacked signal (eg from the top of the cell).

The levels in IF show that ATF3 levels are not decreased in TSSC (while the mRNA is), and accordingly a strong cytosolic ATF3 signal is detectable, the mechanism overseeing this important novelty should be investigated.

• In figure 2A the differences in cal27 cell proliferation after AFT3 deletion, measured with the formazan-forming dye, are slim at best and after day 3 do not increase further, although the cells do not have ATF3. The authors should explain this transient effect. Do the cell die?

• In figure 2E the western blot of Cal27 cells that underwent CRISPR-mediated deletion of ATF3 (with one of the SG guides) shows a band slightly smaller than ATF3 (there are no MW to evaluate the size), but which is still recognised by ATF3 Abs. This is of great concern as it might the result of a recombination/partial deletion leading to a smaller (but functional?) peptide. ATF3 mRNA levels should be tested and the results complemented by the use of ATF3 shRNAs

• From the presented data it is not clear how Interferon-alfa inducible protein 6 and 27 would exert their function(s) towards cell proliferation, transformation etc. More importantly, it is not known if these transcripts are direct targets for ATF3 in TSCC.

Reviewer #4: In this report, Xu et al., has described interesting findings of the tumour suppressor role of ATF3 in oral squamous cell carcinoma with IFI6 and 27 as the prominent downstream targets. The study design is straightforward and the results showed are intriguing.

Major points:

SCCO28 is a highly metastasis-like oral cancer cell line, how would the authors explain the high expression there?

The work is heavily relayed on 2D cell scratching assay, which only represent the migration potential of the cells, while at least cell differentiation has not been measured.

Figure 1. For human oral SCC, please include low magnification for overview of the clinical samples, and multiple samples for each category of the cancer. Please mark epithelial-mesenchymal junctions Panel B labelling is misaligned.

Figure 6. The analysis of tumourigenesis assay are missing, differentiation, proliferation and apoptosis etc. have to be analysed to complete the story. Also are there any metastasis signs? Please consider to remove the photos of mice. For panel B and E, scales are missing. For C and F, the individual sample has to be indicated on the plot.

Minor points:

1. Full names for IFI etc. have to be indicated at the first instance.

2. What do you mean "disrupted localization of ATF3 from the nucleus to cytoplasm"? shouldn't that be opposite?

3. You should acknowledge and cite the original provider for SCC4 and SCCO28.

4. What was the normal oral epithelial culture condition?

5. For puromycin selection on cancer cells, are 48h enough? If yes do you have the control and proof?

6. For antibody (IHC, IF and WB), there are many writing mistakes. Please indicate the antibody full name (such as HRP conjugated etc.) working concentration and company's full name.

7. What is the siRNA concentration?

8. 36B4 should be 36beta4.

9. Which sever has the RNA-seq result been deposited? Which access number?

10. nude/nude mice are which genetic background?

11. "P value" should be "p-value".

12. Molecular weights for the WB bands should be indicated.

**Have all data underlying the figures and results presented in the manuscript been provided?**

Reviewer #1: Yes

Reviewer #2: Yes

Reviewer #3: Yes

Reviewer #4: **No: **RNAseq data deposition and access have not been provided.

PLOS authors have the option to publish the peer review history of their article (what does this mean?). If published, this will include your full peer review and any attached files.

Reviewer #1: No

Reviewer #2: No

Reviewer #3: No

Reviewer #4: No

---

## [Decision Letter · Decision Letter 1]

18 Nov 2020

Dear Dr Wu,

We are pleased to inform you that your manuscript entitled "ATF3 downmodulates its new targets IFI6 and IFI27 to suppress the growth and migration of tongue squamous cell carcinoma cells" has been editorially accepted for publication in PLOS Genetics. Congratulations!

The revised manuscript was seen by the previous reviewers; as you will see, they are positive and enthusiastic.

Yours sincerely,

Gregory S. Barsh

Editor-in-Chief

PLOS Genetics

Peter McKinnon

Section Editor: Cancer Genetics

PLOS Genetics

Comments from the reviewers (if applicable):

Reviewer's Responses to Questions

**Comments to the Authors:**

Reviewer #1: The authors have successfully addressed my concerns from the previous review. I fully recommend the current manuscript for publication.

Reviewer #2: The authors performed a careful revision and answered all my critical comments. The paper has improved tremendously. In my view, the paper is ready for publication.

Reviewer #3: In the resubmitted manuscript Lin Xu and colleagues have included additional experimental evidence and changes to convincingly answer most of the reviewer's scientific and technical concerns. Although the exact mechanism of IFI6 and IFI27 -mediated regulation of TSCC biological effect is still only extrapolated from previous publications in other systems, the manuscript is significantly strengthened and fully supports the Authors' conclusions.

Reviewer #4: I am happy with the revisions that the authors have performed, which have addressed all my questions and suggestions.

**Have all data underlying the figures and results presented in the manuscript been provided?**

Reviewer #1: Yes

Reviewer #2: Yes

Reviewer #3: Yes

Reviewer #4: Yes

PLOS authors have the option to publish the peer review history of their article (what does this mean?). If published, this will include your full peer review and any attached files.

Reviewer #1: No

Reviewer #2: **Yes: **Reinhard Fässler

Reviewer #3: No

Reviewer #4: No

**Data Deposition**

http://datadryad.org/submit?journalID=pgenetics&manu=PGENETICS-D-20-00583R1

**Press Queries**

---

## [Editor Report · Acceptance letter]

28 Jan 2021

PGENETICS-D-20-00583R1 

ATF3 downmodulates its new targets IFI6 and IFI27 to suppress the growth and migration of tongue squamous cell carcinoma cells 

Dear Dr Wu, 

We are pleased to inform you that your manuscript entitled "ATF3 downmodulates its new targets IFI6 and IFI27 to suppress the growth and migration of tongue squamous cell carcinoma cells" has been formally accepted for publication in PLOS Genetics! Your manuscript is now with our production department and you will be notified of the publication date in due course.

With kind regards,

Alice Ellingham

PLOS Genetics

On behalf of:
